# A High-Dimensional Window into the Micro-Environment of Triple Negative Breast Cancer

**DOI:** 10.3390/cancers13020316

**Published:** 2021-01-16

**Authors:** Iris Nederlof, Hugo M. Horlings, Christina Curtis, Marleen Kok

**Affiliations:** 1Department of Tumor Biology and Immunology, The Netherlands Cancer Institute, 1066 CX Amsterdam, The Netherlands; i.nederlof@nki.nl; 2Department of Pathology, The Netherlands Cancer Institute, 1066 CX Amsterdam, The Netherlands; h.horlings@nki.nl; 3Division of Oncology, Department of Medicine, Stanford University School of Medicine, Stanford, CA 94305, USA; ccurtis2@stanford.edu; 4Departments of Medical Oncology and Tumor Biology and Immunology, The Netherlands Cancer Institute, 1066 CX Amsterdam, The Netherlands

**Keywords:** single cell, immune profiling, breast cancer, spatial profiling, tumor evolution

## Abstract

**Simple Summary:**

Triple negative breast cancer (TNBC) shows a substantial level of genomic, cellular, and phenotypic heterogeneity. While genomic heterogeneity and subclonal diversity are prevalent in this subgroup of tumors, a growing body of evidence indicates that the disease course depends on the interaction between cancer cells and the tumor micro-environment (TME). The TME is not static and can change over time, owing to differences in cell numbers, phenotypes, and spatial relationships. Efforts to further elucidate the TME have been aided by a plethora of new technologies that study tumors in a high-dimensional manner. These high-dimensional technologies enable comprehensive analysis of cell phenotypes at the single cell level or the spatial relationships of tumor and immune cells. In this review, we discuss studies in TNBC that unravel specific spatial patterns of cells in the breast TME and single cell phenotypes.

**Abstract:**

Providing effective personalized immunotherapy for triple negative breast cancer (TNBC) patients requires a detailed understanding of the composition of the tumor microenvironment. Both the tumor cell and non-tumor components of TNBC can exhibit tremendous heterogeneity in individual patients and change over time. Delineating cellular phenotypes and spatial topographies associated with distinct immunological states and the impact of chemotherapy will be necessary to optimally time immunotherapy. The clinical successes in immunotherapy have intensified research on the tumor microenvironment, aided by a plethora of high-dimensional technologies to define cellular phenotypes. These high-dimensional technologies include, but are not limited to, single cell RNA sequencing, spatial transcriptomics, T cell repertoire analyses, advanced flow cytometry, imaging mass cytometry, and their integration. In this review, we discuss the cellular phenotypes and spatial patterns of the lymphoid-, myeloid-, and stromal cells in the TNBC microenvironment and the potential value of mapping these features onto tumor cell genotypes.

## 1. Introduction

Tumor heterogeneity is associated with therapy resistance and poor prognosis in a variety of solid tumors [1,2]. Triple negative breast cancer (TNBC), in particular, shows a substantial level of genomic, cellular, and phenotypic heterogeneity [3,4,5,6]. While genomic heterogeneity and subclonal diversity are prevalent in this subgroup of tumors, and accompanied by high-levels of genomic instability, a growing body of evidence indicates that the disease course depends on the interaction between cancer cells and the tumor micro-environment (TME). The TME is not static and can change over time, owing to differences in cell numbers, phenotypes, and spatial relationships. Immune cells, especially cytotoxic T cells, have been the center of attention in view of the rise of immune checkpoint blockade and their potential to kill the tumor cells [7,8,9]. In breast cancer, the endogenous anti-cancer immune response is often expressed as the level of tumor infiltrating lymphocytes (TILs) and is tightly associated with prognosis and response to (immuno-)therapy in TNBC [10,11,12,13,14,15,16,17,18]. However, a low level of TILs does not equate to disease progression. As the response rates to anti-PD-(L)1 therapy in metastatic TNBC and the combination of anti-PD-(L)1 and chemotherapy in primary TNBC have been modest [14,15,16,17,18], there is a clinical need to understand why the majority of the patients remain without an effective response. Thus, further characterization of the TME may provide a biological rationale for novel immunomodulatory strategies.

Efforts to further elucidate the TME have been aided by a plethora of new technologies that study tumors in a high-dimensional manner. These high-dimensional technologies enable comprehensive analysis of cell phenotypes at the single cell level or the spatial relationships of tumor and immune cells. High-dimensional phenotyping of the breast TME has been successfully achieved by technologies like flow cytometry, single cell mass cytometry, and single cell RNA sequencing (scRNAseq) [19,20,21,22,23] (Table 1). Technologies that preserve the spatial relationships between cells in TNBC include multiplex immunofluorescence, imaging mass cytometry (IMC) and multiplex ion beam imaging (MIBI) [24,25,26,27] (Table 1). Most studies do not capture the dynamics of the TME yet, as it requires sequential tissue biopsies which are difficult to obtain in patients. Nevertheless, information on the evolutionary path of breast tumor cells in the context of their TME can potentially guide the design of synergistic immunotherapy combinations for relatively cold tumors like breast cancer.

In this review, we discuss studies in TNBC, and to a lesser extend in other breast cancer subtypes, which unravel specific spatial patterns of cells in the breast TME and single cell phenotypes. In particular, we consider the phenotypes and patterns of the lymphoid cells, myeloid cells, and stromal cells in the local breast cancer TME.

## 2. Lymphoid Spatial Phenotypes

The spatial pattern of lymphocytes in the TME is often exemplified by three phenotypes: the immune-inflamed, the immune-excluded, and the immune-deserted tumors [41]. Although TNBC patients usually have higher degrees of immune infiltration compared to hormone receptor positive patients [11], poorly-infiltrated phenotypes are no exception in TNBC (~12 to 42%) [24,25]. Poorly infiltrated breast tumors may exhibit tumor-intrinsic properties that contribute to the successful evasion of immune detection for example via a low number of neoantigens or downregulation of elements of the antigen presentation machinery [9]. Both the immune-deserted and immune-excluded phenotype are characterized by the restricted accumulation of lymphocytes in the tumor, albeit the immune-excluded tumors (~31% TNBC reported by [25]) show a high presence of immune cells localized at the border that fail to invade the tumor bed [41]. Characterization of the immune border of breast tumors revealed a loss of HLA-1 and an increase in B7-H4 immune regulators, impeding antigen recognition by T cells, thus suggesting effective immune evasion [25]. Moreover, the expression of TGF-β has been implicated in the exclusion of T cells [42]. A summary of the mechanisms and markers that lead or are correlated to poor infiltration are illustrated in Figure 1A (left-side). 

The classification of immune excluded tumors remains a challenge. While a study of 38 TNBCs with CD8^+^ directed immunohistochemistry reported 12 immune-excluded ("margin restricted") tumors, a study of 41 TNBCs with MIBI only reported "cold" tumors and made no separate category for immune-excluded tumors [24]. Whether or not immune-excluded tumors in this study are absorbed in inflamed or immune-deserted tumors and if this is due to examination of limited regions of tissue (individual cores on a tissue microarray, TMA), remains unclear. Most TMAs are preselected for tissue containing tumor cells and may obscure the immune-excluded phenotype as immune cells are restricted to the outer tumor margin.

The inflamed phenotype is characterized by the presence of lymphoid immune cells in the tumor [41]. In breast cancer, most studies have largely confirmed the existence of this inflamed phenotype but divide those usually in two subclasses. The first inflamed subclass harbors infiltration with T cells both Stromal and Intratumoral; inflamed-SI. The second inflamed subclass harbors infiltration with T cells restricted in mainly the stroma in the tumor area, albeit not restricted to the outer tumor margin; inflamed-Stroma Restricted (inflamed-SR) [24,25,26,27,39]. This division of inflamed tumors results in a total of four spatial lymphoid phenotypes (Figure 1B). It is important to realize that inflamed-SI and inflamed-SR tumors consist of similar (high) numbers of lymphoid cells but differ in their spatial organization and that the prevalence of the inflamed-SI and inflamed-SR spatial phenotypes within TNBC are equal [24,25]. Gruosso and colleagues described these spatial patterns with the localization of CD8^+^ T cells in 38 therapy-naïve TNBC samples using immunohistochemical analysis [25]. The existence of the inflamed-SR and inflamed-SI TNBC tumors was additionally confirmed with highly multiplexed technologies including IMC and MIBI. The 59 basal-like tumors that were analyzed with IMC using a tumor-focused panel with few immune markers showed a high level of same-cell contact among both epithelial and stromal cells compared to other breast cancer subtypes, indicating a separation of compartments [39]. The analyses of 41 TNBC with MIBI using a highly immune focused antibody panel also showed that regions in the tumor can either be mixed or comprised of either predominantly immune or tumor cells [24]. Inflamed-SI TNBC show more inflammatory signaling (e.g., interferon-γ, cytotoxins [25]), whilst inflamed-SR TNBC shows distinct immunosuppressive profiles. A summary of the mechanisms and markers that lead or are correlated to inflamed tumors are illustrated in Figure 1A (right-side). The prognostic value of these spatial patterns in TNBC is still under discussion as studies have reported contradicting results for a better survival with the inflamed-SR phenotype [24] or the inflamed-SI phenotype [25].

Additionally, pinpointing the difference between the immune-excluded and immune-SR phenotype is challenging, as it is currently uncertain whether or not the immune-excluded phenotype is a truly distinct biological phenotype or actually similar to inflamed-SR with, e.g., a lower amount intratumorally dispersed stroma or a higher level of exclusion and restriction processes as illustrated in Figure 1A. Furthermore, the often-smaller areas of tissue that are analyzed with multiplexed techniques could potentially lead to a sampling bias and could conceal phenotypes and spatial diversity within a tumor.

## 3. Lymphoid Cell Phenotypes

Single cell profiling of breast tumors reveals a multitude of T cell and B cell phenotypes [19,20,21,29,43]. Effector memory T cells (T_EM_; CD45RO^+^CD45RA^−^CCR7^−^ T cells) are the most prevalent T cells in all breast tumors [19,21,23,27,43], indicating that the majority of the T cells in the tumor is antigen experienced. CD4^+^ T cells, CD8^+^ T cells, and CD20^+^ B cells are more abundant in all breast cancer subtypes compared to normal breast tissue. Below we describe the phenotypes and spatial position of CD8^+^ (effector) T cells, CD4^+^ T cells, regulatory T cells, B cells and natural killer cells within the TME of breast tumors with a focus on TNBC.

### 3.1. CD8^+^ T Cells

The majority of CD8^+^ T cells in all breast tumor subtypes are T_EM_ (CD45RA^−^CCR7^−^; 72%), accompanied by a smaller group of T_EMRA_ (effector memory CD45RA^+^, CD45RA^+^ CCR7^−^; 23%), and minor populations of T_CM_ (central memory, CD45RA^−^CCR7^+^; 3%), and T_NAIVE_ (CD45RA^+^CCR7^+^;1%) [23]. It is largely unknown if CD8^+^ effector T cells exhibit a different phenotype in inflamed-SI tumors where the distance between tumor cell and T cells is lower compared to inflamed-SR tumor cells.

Several basic immunology studies have described a population of memory T cells that are settled within tissue and do not recirculate [44,45,46]. These CD8^+^ T cells are called tissue-resident memory T cells (T_RM_) and are characterized by the upregulation of CD103 and CD69 [47]. Both CD103 and CD69 limit T cell tissue egress by downregulation of several chemotaxis regulators (e.g., S1RP1 and KLF2) and receptors (KLRG1) and are crucial determinants of T_RM_ retention in epithelial tissues, including breast tumors. [23,48,49,50]. A substantial proportion of the CD8^+^ T cells in breast tumors are T_RM_ (38%; [23]), defined by the expression of CD103 and CD69 [23,27]. In murine cancer models, it was discovered that CD69 and CD103 on T cells act in a sequential nature, where CD69 is crucial for the entering of the tissue and CD103 for the persistence of the cell in the tissue [51]. This functional sequence of infiltration is also observed in TNBC and linked to their spatial patterns. CD69^+^CD103^+^ T cells (T_RM_) are highly enriched close to triple negative tumor cells (e.g., inflamed-SI), CD69^+^CD103^−^ CD8^+^ T cells are more evenly distributed between stroma and cancer cell islands (e.g., both inflamed-SI and inflamed-SR), and CD69^−^CD103^−^CD8 T cells are found almost exclusively in the stroma (e.g., inflamed-SR) [27] (Figure 1). T_RM_ show enhanced cytotoxic signaling (e.g., GZMB and PRF1) and tumor cell killing potential [23,38,47,52,53]. Cytotoxic T cells in the inflamed-SI TNBC showed higher granzyme B (GZMB) expression than inflamed-SR TNBC [25], suggestive of T_RM_ cell phenotype. In addition to the enhanced cytotoxic profile, an increase in the expression of immune-checkpoint and antigen presentation is also observed in the T_RM_, including the expression of HLA-DR [20], and immune-checkpoints PD-1, CTLA-4, TIGIT, TIM3, and LAG3 [23,27], in line with previous characterizations of T_RM_ in other cancer subtypes [53].

The T_RM_ phenotype in breast cancer with a high expression of co-inhibitory receptors and high expression of GZMB is remarkably similar to the phenotype attributed to late dysfunctional (or "exhausted") T cells [54,55,56], suggesting that T_RM_ in breast cancer are dysfunctional. In literature, T cell dysfunction is described on a scale from predysfunctional T cell states that transition into late dysfunctional T cells, where dysfunctional T cells are considered to have been persistently triggered by their tumor antigens and are which are sculpted by immunomodulatory signals in the TME [54]. Single cell sequencing technologies in breast cancer characterize T cell phenotypes that are largely consistent with the early and late dysfunctional profiles, although they are not named as such. [21]. Specific T cell clusters in the breast cancer TME of all subtypes [21,29] show differential expression for predysfunctional markers (e.g., TCF7), "transitional markers" (e.g., GZMK, IL7R, and PD-1) and late dysfunctional markers (PD-1, CTLA-4, TIM3, and LAG3). While PD-1 expression is mainly attributed to dysfunctional (exhausted) T cells in lung cancer [55], PD-1 expression in breast cancer was detected in higher frequencies in both CD103^+^CD69^+^ and CD103^−^CD69^+^ T cells compared to CD103^−^CD69^−^ T cells [27], suggesting that PD-1 upregulation may be one if the first steps in T_RM_ differentiation and effector T cell (pre)dysfunctionality in breast cancer.

The phenotype of T_RM_ and their transition in late dysfunctional profiles make them intriguing candidates to exploit in the anti-cancer immune response of patients with breast cancer. Indeed, the presence of T_RM_ cells in the breast cancer TME (assessed by a single cell RNA derived signature) was associated with favorable overall and relapse-free survival in primary TNBC [23], as well as with response to pembrolizumab in patients with metastatic TNBC (KEYNOTE-086) [57]. However, to the best of our knowledge, preselection of breast cancer patients with T_RM_ immune profiles is currently not incorporated in any clinical trial with (immune) therapeutical interventions.

### 3.2. CD4^+^ T Cells

CD4^+^ T cells also infiltrate TNBC (~15%, similar to CD8^+^ T cells) with similar patterns in inflamed-SR and inflamed-SI tumors [24]. The proportional increase of CD4^+^ T cells in all breast cancers compared to normal tissue is higher than the proportional increase of CD8^+^ T cells [20,33,43,58], consistent with their central role in maintaining the delicate balance between protective and inflammatory immune responses in the tumor. However, when total TIL levels increase, the CD4^+^/CD8^+^ TIL ratio decreases, suggesting that it is not the CD4^+^ T cell population that must expand for an effective immune response [23]. The majority of the CD4^+^ T cells in all breast tumors are T_EM_ (CD45RA^−^CCR7^−^; 88%), followed by a small group of T_EMRA_ (CD45RA^+^CCR7^−^; 8%), T_CM_ (CD45RA^−^CCR7^+^;4%), and T_NAIVE_ (CD45RA^+^CCR7^+^; 1%) [23]. Provided that the different phenotypes of CD4^+^ T cells are linked to a distinct function and interactions with cell types, distinct spatial patterns of CD4^+^ T cells would be expected. scRNAseq of CD4^+^ T cells from several breast cancer subtypes reported that CD4^+^ T_EM_ and T_CM_ clusters exhibit variable levels of gene expression involved in type I and II interferon response, hypoxia, and anergy, indeed indicating a different signaling and role in the tumor [21]. However, there is only scarce information on the spatial position of CD4^+^ T cells and their clinical relevance. The T_RM_ phenotype (CD103^+^) —that is linked to intratumoral infiltration in CD8^+^ T cells— is also observed for CD4^+^ T cells, although at significantly lower number [23]. What specific role and colocalization these CD4^+^ T_RM_ might have in breast cancer and if their presence favors a good response to (immuno-)therapy is still unclear. CD4^+^ T cells express high levels of PD-1 (similar to CD8^+^ T cells), and even higher levels of cytotoxic T lymphocyte-associated protein-4 (CTLA-4) compared to CD8^+^ T cells [23,24]. The spatial analysis of TNBC with MIBI reported that some patients have predominantly PD-1^+^CD4^+^ T-cells whereas others have predominantly PD-1^+^CD8^+^ T-cells [24], yet whether this leads to a different anti-cancer response is unclear.

CD4^+^ T cells in highly infiltrated breast tumors of all subtypes are also phenotypically skewed towards the T follicular helper cells (T_FH_) phenotype, characterized by CD200, ICOS, CXCL13, and a high level of PD-1 [33]. T_FH_ constitute 40% of the PD-1^+^ CD4^+^ TIL in breast cancer [43], suggesting a prominent role in the anti-breast cancer immune response. Interestingly, the proportion of T_FH_ cells is not correlated with the abundance of PD-1^+^CD8^+^ TIL, suggesting (partial) non-overlapping immune modulatory mechanisms. As implied by their name, T_FH_ are localized in follicular structures, in breast cancer namely the tertiary lymphoid structures (TLS) [34]. TLS are frequently present in primary breast tumors [59,60] and may signal local immune responses directed to the tumor, as suggested by their prognostic value in breast cancer [61,62] and other malignancies [63,64]. The variety of immune checkpoints in TLS [65] and networks of dendritic cells [34,66] are additional clues for their role in local immunity. In melanoma, tumors without TLS have a dysfunctional molecular phenotype, suggesting a link between TLS and CD8^+^ T cells [64].

### 3.3. B Cells

Several classes of B cells can be identified in all breast tumor subtypes, including naïve and memory B cells, centroblasts, and germinal center B cells [29,34,43]. In the majority of the breast tumors (~80%), B cells are present at relatively low levels in the tumor (around 2–3%), yet they are more abundant in the tumor than in normal breast tissue [34]. In addition, B cells are more abundant in hormone receptor negative tumors [34]. When TLS are present around the tumor, B cell levels are higher in the peritumoral area. The localization of B cells has therefore mainly been studied in the context of TLS: B cells are most frequently detected in TLS and when a TLS is present in the tumor, this leads to relatively more germinal center (GC) B cells [34]. It is currently unknown if the B cells infiltrated between the breast tumor cells have a significant different phenotype or function as opposed to the B cells in the TLS around the breast tumor. B cells’ diverse roles in humoral immunity, antigen presentation, modulation of T cells, and innate immune cells [67] suggest that there are various biological relevant spatial patterns and colocalizations with different cell types that yet have to be uncovered. For example, studies using MIBI revealed that B cells are consistently depleted along the tumor border of TNBCs [24], yet the cause of this relative depletion was not uncovered.

The level of B cells is well correlated with TILs levels [34], though their presence harbors independent prognostic information for TNBC and HER2^+^ breast cancer. The potential of using immunoglobulin repertoires to study the anti-tumor response or to use as biomarkers to predict the efficacy of (immuno)therapeutic interventions [67] can be potentially uncovered with future single cell RNA-based immunoglobulin repertoire studies.

### 3.4. Regulatory T cells

Regulatory T cells (T_REG_ cells), defined by the expression of transcription factor Foxp3 [68,69], have a central role in maintaining immune tolerance throughout the body. Although the level of T_REG_ cells in all breast cancer subtypes is low [23,58], their role is deemed important for anti-cancer immunity. In hormone receptor positive breast cancer, patients with clinical benefit from combination targeted therapy (tamoxifen/vorinostat/pembrolizumab) showed a prominent treatment induced depletion of (CD4^+^Foxp3^+^CTLA4^+^) T_REG_ [70]. It is thought that T_REG_ cells can facilitate tumor growth and metastasis based on the observed regression of established tumors and prevention of metastasis development in experimental models of T_REG_ cell depletion [71,72].

With the use of high-dimensional technologies, different populations of T_REG_ cells are identified in breast tumors of all subtypes [21,23]. In summary, T_REG_ cell clusters in breast tumors usually have a (selection of) well-activated phenotype with high expression of immune checkpoint molecules (CTLA-4, TIGIT, GITR) [21], expression of cytotoxic molecules [73], and high expression of the chemokine receptor CCR8 [23,58]. Breast cancer associated T_REG_ cells show a significantly higher expression of CTLA-4 and tumor necrosis factor receptor superfamily, member 4 (TNFRSF-4, also known as OX-40) on their membranes compared to other CD4^+^ and CD8^+^ T cells [23,58]. CTLA-4 expressed by T_REG_ cells impairs maturation of APCs, such as DCs, by binding to CD80/86 [74] or by transendocytosis of CD80/86, which then inhibits CD28-mediated costimulation of T cells [75]. The co-expression of CTLA-4 and CD80 was indeed observed in one T_REG_ cluster in the breast cancer TME [21]. scRNAseq also identified separate T_REG_ clusters with an enrichment for inducible T-cell costimulator (ICOS) compared to the other T_REG_ clusters [21]. ICOS is involved in the proliferation of activated T_REG_ cells through binding to an ICOS ligand expressed by plasmacytoid DCs [76]. From the single cell transcriptional profiles of T_REG_ in breast cancer it is unclear if the presence or proportion of different T_REG_ clusters have clinical consequences.

### 3.5. Natural Killer Cells

Natural killer (NK) cells are cytotoxic innate lymphoid cells that produce proinflammatory molecules and are reported to be present at low frequency (<1%–5%) in breast tumors of all subtypes with considerable interpatient variation [19,24,33]. A recent pan-cancer scRNAseq study identified two NK cell clusters, including one cytotoxic NK cell cluster and one NK cell cluster involved in dendritic cell chemo-attracting [56]. In breast cancer, up to nine different NK clusters were identified with scRNAseq (potentially including NKT cells) [21]. However, despite picking up the signal of NK cells in breast tumors, few studies have elaborated on their spatial localization and clinical relevance. Moreover, the differences or similarities of NK cells between TNBC and other breast cancer subtypes is still unknown. By blocking the inhibitory NKG2A receptor in head and neck cancer as a prelude [77,78], new results on targeting and exploiting NK cell activity in breast cancer patients (e.g., CT04307329) are expected.

## 4. Myeloid Cell Phenotypes and Spatial Patterns

Despite being a common infiltrating cell type in breast cancer, myeloid cells have received less attention compared to lymphoid cells. Nevertheless, several phenotypes for macrophages and other myeloid cells have been described and are discussed in this section. Macrophages are a diverse group of cells and are important modulator and effector cells in the immune response. Macrophages in tumors are termed tumor associated macrophages (TAMs) and are often abundantly present in breast cancer. TAMs facilitate angiogenesis and remodeling of the extracellular matrix in preclinical models, supporting progression of disease [79]. In addition, TAMs have been implicated in the restriction of T cells in the micro-environment [80] and blocking TAMs recruitment in breast cancer mouse models can increase CD8^+^ T cell infiltration and their cytotoxic activity in the primary tumor [81,82]. Traditionally, macrophages are grouped in two polarized macrophage clusters, termed M1 and M2 [83,84]. However, categorizing TAMs as either M1 or M2 is difficult as TAMs may not form clear-cut activation subsets nor expand clonally like T cells. Indeed, single cell analyses of breast cancer show several monocytic (macrophage) clusters with different activation modus status [21,24]. Both M1 and M2-like phenotypes were identified, and their presence showed a high correlation with each other, suggesting that M1 and M2 macrophages coexists and may not be polarized [21]. The suppressive nature of TAMs is illustrated by the substantial subset of macrophages in breast tumors that express PD-L1, IDO and other immune checkpoints [19,24,85].

Myeloid cells also show patterns in the context of the lymphoid distribution in TNBC (Figure 1). Inflamed-SI TNBC show an increase in proinflammatory macrophages (CD68^+^CD206^+^) compared to inflamed-SR tumors [25]. Moreover, at the border of inflamed-SR TNBC, immunosuppressive PD-L1, PD-1 and IDO was found on CD11c^+^CD11b^+^ immune cells, suggestive of myeloid cells [24]. In contrast, inflamed-SI tumors showed overall higher levels of immune checkpoints [25], with PD-L1 and IDO protein expression mainly on tumor cells and PD-1 protein expression mainly on CD8 T cells [24], suggesting that the role of myeloid cells diverges with different lymphoid patterns. TNBC patients treated with neoadjuvant chemotherapy and PD-1 checkpoint inhibition with a pCR expressed higher PD-L1 expression on both tumor cells and macrophages compared to patients with a non-pCR [86]. As of yet, it is unknown which PD-L1 expressing cells are important for PD-1 blockade efficacy. Future single cell studies will probably provide important information on this topic.

There are very few multiplexed proteomic data on other myeloid cells in the breast TME. Using gene expression analysis and immunohistofluoresence an enrichment for IL-17 producing cells and neutrophils was detected in inflamed-SR TNBCs [25]. Typically, such cells are localized near the tumor border. Both the presence of neutrophils and eosinophils have been reported in single cell sequencing studies in breast cancer studies [21,24], but further details are scant. In summary, most high-dimensional studies on the human breast cancer TME have overlooked myeloid cells, despite their ample presence in the tissue and immune suppressive capacities. A potential explanation may be the higher vulnerability of myeloid cells compared to lymphoid cells during sample preparation, resulting in an underrepresentation of this cell type in most comprehensive studies.

## 5. Stromal Cell Phenotypes and Spatial Characteristics

The composition of TNBC includes many other stromal cells in addition to the tumor-, lymphoid- and myeloid cells. Stromal factors that can contribute to altered infiltration in breast tumors include tumor associated fibroblast infiltration and oxygen homeostasis.

### 5.1. Tumor Associated Fibroblasts

The role of fibroblasts in breast tumor development and progression was already described over two decades ago, when the wound healing response of fibroblasts was shown to be frequently activated in breast cancer and a predictor of clinical outcome in patients with early-stage breast cancers [87]. Nevertheless, the presence of fibroblasts lacks discriminative power within TNBC since almost all basal-like samples (42/45) showed an activated wound healing response [87,88]. New single cell technologies have enabled the investigation of fibroblasts with distinct origins and functions in multiple cancer types [22,28,56]. A recent pan-cancer scRNAseq study showed that fibroblasts are highly versatile cell types endowed with extensive heterogeneity and cancer-type recognition [56]. In breast cancer several fibroblasts population with distinct functions have been identified. For example, two fibroblast subsets (CAF-S1, CAF-S4) are common in TNBC. Predominant infiltration of the immunosuppressive fibroblast (CAF-S1) is linked to increased T lymphocyte survival, increased T_REG_ differentiation and inhibition of effector T cell proliferation [22]. The immunosuppressive properties of fibroblasts are often attributed to their role in remodeling the extracellular matrix [89] and the synthesis of molecules that deploy immunosuppressive signals in the environment [42,90] (e.g., via expression of TGF-β). Single cell RNA sequencing of the CAF-S1 subset revealed up to 8 clusters of immunosuppressive CAF-S1 in breast cancer [28]. Especially the abundance of CAF-S1 subtypes characterized by high expression of genes coding extracellular matrix (ECM) proteins and TGF-β signaling pathway were associated with an immunosuppressive environment enriched in T_REG_s and anticorrelated with CD8^+^ T cells. TGF-β expression is tightly linked to exclusion of T cells from the tumor core in a murine breast cancer model and in patients with renal cell carcinoma [42]. This is in line with the observation that poorly infiltrated TNBC show an enrichment of gene expression for fibrosis and TGF-β [25]. However, differential expression in this study was not separately done for immune-excluded and immune-deserted tumors, probably due to low sample size of the immune-deserted tumors. As such, it is unclear whether TGF-β signaling is a property of poorly-infiltrated tumors, or immune excluded tumors. Spatial analysis with IMC showed that when the TME was enriched for fibroblasts, there were few interactions between immune cells [39], in accordance with their role in immune exclusion. The anti-correlation between CD8^+^ T cells and the immunosuppressive CAF-S1 fibroblast subtype was particularly high in TNBC [22]. In summary, breast tumors can span a spectrum from immune cell dominant to fibroblast dominant tumors [26,39] that drives immune exclusion mechanisms [25,42] and depends on the subtype of fibroblasts in TNBC [22] (Figure 1).

### 5.2. Oxygen Homeostasis: Hypoxia and Angiogenesis

Both hypoxia and angiogenesis are associated with the composition of the TME and immune cell function. Based on IMC analysis, breast tumor cells of all subtypes that expressed carbonic anhydrase IX, a marker of hypoxia, were found to be associated with genomic gains of PD-L1, and heterozygous deletions of β2-microglobulin, suggesting that tumor cell hypoxia is associated with genomic alterations that facilitate immune evasion and suppression [39]. It remains unclear whether these genomic alterations lead to TME hypoxia or vice versa. In addition, correlation between hypoxia and T cell states has been described [21]. A hypoxic tumor environment promotes the production of pro-angiogenic factors [91,92], including TGF-β (immunomodulation described above) and vascular endothelial growth factor (VEGF), leading to neo-vascularization. In preclinical models, VEGF receptor inhibitors showed immunomodulatory effects, including enhanced tumor infiltration of immune cells, and reduced immunosuppressive effects of myeloid cells [93]. The combination of immune checkpoint inhibition with a VEGF-targeted antiangiogenic therapy in renal cell carcinoma led to significant longer progression-free survival.

The localization of T cells and proliferating immune cells in all breast cancer subtypes is observed in the vicinity of vascular endothelium cells [24,26,39], indicating that T cell presence may be dependent on the vascular endothelium of breast tumors. Moreover, vascular endothelial cells in tumors can impede extravasation of immune cells by altered expression of leukocyte adhesion molecules and FAS-ligand [94,95,96,97,98]. The expression of FAS-ligand was also observed in breast cancer (n = 62) with a large variability between samples [98]. The impaired extravasation in tumors may be cell-specific as T_REG_s still could extravasate where CD8^+^ T cells could not [98]. The process of T cell restriction within the tumor bed e.g., via altered extravasation in TNBC remains unclear.

## 6. Integrating Tumor Micro-Environmental Features and Genomic Heterogeneity

To date, tumor heterogeneity has primarily been defined based on genetic diversity because genome sequencing, which is now commonplace, naturally reveals such patterns. For example, by tracing the evolutionary history of tumor cells through bulk, multi-region, and single cell DNA-sequencing it is possible to identify clones that persist or expand under therapeutic pressure or seed metastatic sites [99,100,101,102,103], as depicted in Figure 2A. The distinct evolutionary paths of tumor clones manifest as spatial variations that are observed in several multiregional sequencing studies in solid tumors [104,105]. A multiregional whole-exome sequencing study in HER2 positive breast cancer revealed that if two biopsies are obtained randomly from a primary breast tumor at diagnosis, more than a quarter of the clonal mutations observed in one biopsy would be absent in the other simply due to the heterogeneity present prior to treatment [99]. Significant spatial heterogeneity of point mutations and copy-number changes have also been observed for HER2-negative breast cancer [106]. This suggests a non-random spatial organization of tumor clones in breast cancer. While currently multi-region and single cell sequencing approaches are used, these provide limited spatial resolution. Hence, we cannot fully resolve this topic with the tools at hand. Ultimately, the field will greatly benefit from scalable spatial genomics approaches.

In addition, it is unknown to what extent spatial genetic variation in cancer cells drives tumor progression or are the consequence of spatial constraints, including those imposed by the surrounding TME. The interaction between tumor cells and the TME is crucial for tumor clone persistence and disease progression [107], as tumor growth can be driven by a minor cell subpopulation, which enhances the proliferation of all cells within a tumor by overcoming tumor micro-environmental constraints [108]. Future research should therefore address the relationship between subclonal spatial heterogeneity and interactions with the surrounding TME and study tumor heterogeneity as specific ecological niches, adapted to the composition of tumor clones and specific cells in the TME (Figure 2B). In lung cancer, the T cell repertoire differs across tumor regions and correlates with localized mutations (presumably different clones) [109]. Other studies have found that more shared subclonal mutations are observed between immune cold regions compared to immune hot regions [110]. These studies suggest that the spatial immune cell and tumor cell heterogeneity can be linked.

In TNBC, single cell RNA sequencing revealed a shared malignant tumor cell population with enrichment of genes involved in innate immune sensing and inflammation [6], suggesting interactions with the TME. Spatial proteomic analysis of TNBC via MIBI indicated that p53-positive tumor cells were mostly localized near immune cells in a subset of patients with tumor cells at the border of the stromal and tumor cell compartments were more transcriptionally active (identified by a different methylation pattern) compared to (non-necrotic) the tumor cells located in the core [24]. These observations suggest a spatial interdependence between tumor and immune cells, but more in-depth analyses are required to study these interactions, particularly in the context of therapy.

## 7. Ecological and Evolutionary Dynamics during TNBC Progression

We have so far discussed the use of high-dimensional data to reveal the interplay between micro-environmental and tumor cell heterogeneity. Lastly, we discuss ecological (niches) and evolutionary (population) dynamics during major transitions, including the treatment and colonization of distant metastatic sites (Figure 2B). While tumor progression had traditionally been considered a linear process, whereby increasingly genetically advanced clones eventually acquire metastatic competence, genome sequencing studies have reconstructed the evolutionary path of primary tumors and matched metastasis, revealing that metastatic lineages commonly diverged early [111,112]. Indeed, computational modeling revealed that dissemination commonly occurred before the primary tumor was clinically detectable [111,112]. Along these lines, single-nucleus sequencing of one thousand TNBC cells suggests that the majority of genetic aberrations are acquired at the earliest stages of tumor evolution [113]. Another recent study with paired primary and metastatic lesions, estimated metastatic seeding to occur 2–4 years prior to diagnosis of the primary tumor in breast cancer [106]. Further, the authors found that in the absence of treatment, metastases often arise from the major clone in the primary tumor and lack metastasis-specific drivers [106]. These results are consistent with the finding that driver gene heterogeneity is minimal among untreated metastases [114].

Treatment dramatically remodels clonal architecture and can mutagenize cancer genomes [106,115]. The stringent selective pressures that therapies impose can permit the outgrowth of resistant subclones, that previously represented a minority of the population. Deep exome-sequencing of 20 TNBC patients sampled during the course of therapy revealed clonal extinction in some patients and clonal persistence in others [103]. Remarkably, clonal persistence during chemotherapy was associated with gene signatures that are largely attributed to the entire TME, including extracellular matrix (ECM) degradation and hypoxia [103], indicating that clonal shifts in TNBC are tightly linked to changes of ecological niches. Across breast cancer subtypes, chemotherapy and immunotherapy have been reported to lead to a more cytotoxic T cell repertoire and clonal expansions [16,20,116]. In addition to changes in lymphoid populations, changes in macrophage populations can be observed after chemotherapy [20]. This illustrates that therapy not only affects tumor cells, but also the TME. In addition to the effect of therapy, changes in the TME may contribute to more aggressive phenotypes in metastatic lesions, where lower infiltration of TIL and CD8^+^ T cells and a lower expression of PD-L1 is observed relative to primary tumors [23,30].

Beyond differences in the abundance of cell types, little is known about changes in the spatial topography of the cellular ecosystem in primary tumors versus metastases. When the association between stroma and TILs is investigated in low dimensions throughout tumor evolution (Figure 2A), the proportions of cells within different compartments can be uncovered, but detailed cell-cell interactions lack. Future studies should investigate the relationship between tumor cells and the TME across different niches aiming at uncovering cell–cell interactions and functional heterogeneity (Figure 2B). Tumor clones may migrate to a distant site where they form a new tumor niche that is less sensitive to therapy due to either new genetic alterations or an altered TME (Figure 2).

## 8. Clinical Implementation of TME Profiling

Up to this point, this review has summarized the multitude of cellular phenotypes and spatial patterns of the TNBC TME (Section 2, Section 3 and Section 4) and the necessity to study this in tandem with tumor cell evolution (Section 6 and Section 7). Given the complexity and costs of high-dimensional techniques, translating knowledge gained by comprehensive analyses of the cancer-immune interactions directly into the clinic will be challenging. Where exploratory studies often focus on multiple breast cancer subtypes or stages, clinical trials usually target a specific patient population with similar disease characteristics. The pioneering work of studies that have explored the TNBC TME extensively [19,21,24,25,26,39], can provide rationale for early phase clinical trials that evaluate combination immunotherapy or novel immunomodulatory strategies. A fine-tuned TME assessment of patients treated in the context of a clinical trial with new high-dimensional techniques can potentially elucidate which cell types are driving, supporting, and counter-acting a therapeutic intervention, in both responders and non-responders. We and others [16,117] have previously shown that sequential biopsies in breast cancer patients can aid in the understanding why and how some therapies may induce a more favorable tumor microenvironment. Integrating this approach with even more high-dimensional techniques (as currently done in several trials [38,118]) can teach us not only if and how our target cells, e.g., CD8^+^ T cells, respond, but also how interacting cells, e.g., fibroblasts or macrophages (Figure 1A), adapt. This can potentially uncover which other cells or mechanisms should be stimulated or inhibited in new or combinatorial therapy regimens.

## 9. Conclusions

High dimensional technologies have yielded novel insights into the makeup of the TME within TNBC. The spatial composition of the TME in breast cancer is often expressed as one of four spatial phenotypes, depending on the localization of lymphoid immune cells. However, this approach often overlooks macrophages, NK cells, or fibroblasts that show great immunomodulatory and inflammatory potential. As technologies to enable spatially-resolved single cell profiling at the transcriptomic and proteomic levels advance, this fuels our further understanding of cell-cell interactions without doubt. In this frame, one of the outstanding challenges is to capture transient cell-cell interactions. Future research should also focus on the development of robust methods to integrate tumor cell and TME phenotypes and to map the evolutionary path of the tumor and TME in patients. Exploring the TME within models of cancer heterogeneity will enhance our insight in tumor biology and evolution. In addition, approaching the composition of TNBC as a mix of niches with interacting cells may inform new therapeutic strategies. Moreover, high-dimensional technologies will be instrumental in the quest to discover biomarkers to select patients for therapy and to uncover (targetable) weaknesses of TNBC.

## Figures and Tables

**Figure 1 cancers-13-00316-f001:**
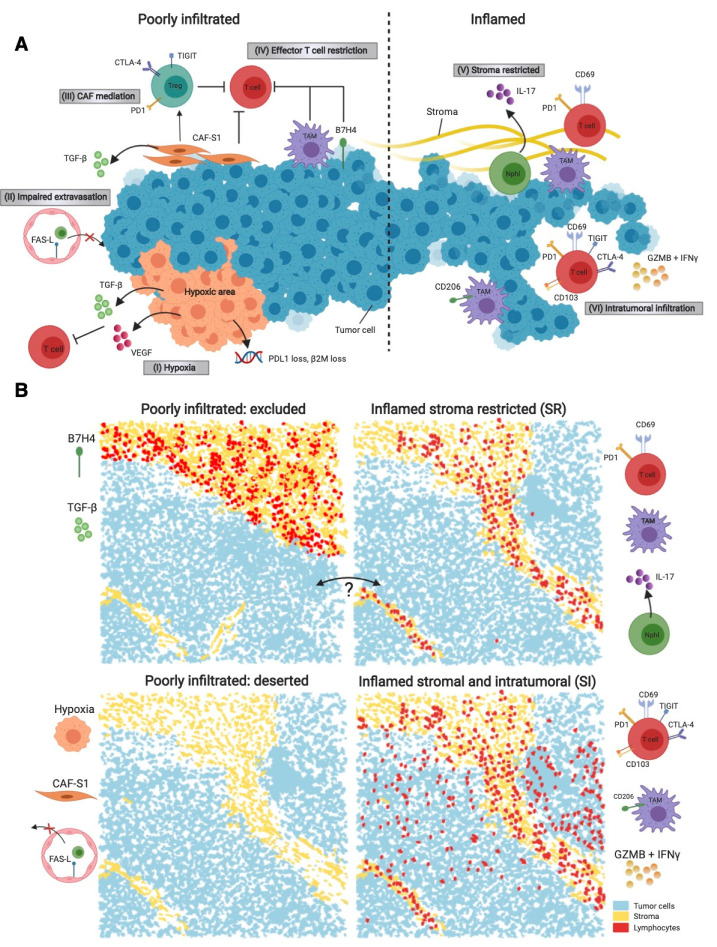
The spatial lymphocytic phenotypes of immune infiltration in triple negative breast cancer (TNBC). (**A**) A schematic illustration of a poorly-infiltrated (left-side) and inflamed (right-side) tumor with key markers and processes. (I) Hypoxia in the tumor promotes the production of pro-angiogenic factors, including TGF-β and vascular endothelial growth factor (VEGF), which can both modulate and inhibit effector T cells; (II) Low infiltration of lymphocytes occurs due to their impaired extravasation, e.g., through endothelial expression of FAS-ligand (FAS-L); (III) CAF mediation, e.g., via CAF-S1 fibroblasts, inhibit T cells directly and via the modulation of regulatory T cells; (IV) Effector T cell restriction through tumor associated macrophages (TAMs) and immuno-inhibitory molecules, such B7H4. B7H4 can be expressed on e.g., TAMs or tumor cells to inhibit effector T cells; (V) When T cells are present mainly in the stroma, this is characterized by the presence of TAMs without the expression of CD206 and neutrophils that express IL-17. Phenotypically, the stroma-restricted T cells express PD-1 and CD69; (VI) When tumors harbor also intratumoral infiltration, this is accompanied by activated TAMs with CD206 expression, inflammatory markers like granzyme B (GZMB) and interferon-g (IFN γ). Phenotypically, the intratumoral T cells express CD69, CD103 and multiple immune checkpoints, including PD-1, CTLA-4, and TIGIT. (**B**) In TNBC, tumors are often divided in poorly infiltrated (excluded and immune deserted, left panels) and inflamed (stroma restricted (SR) or stromal+intratumoral (SI), right panels). "Immune excluded" tumors show a lack of lymphocytes in the tumor, but lymphocytes are present at the invasive margin. Characteristics: Expression of TGF-β and immuno-inhibitory molecules, such as B7H4 exclude the immune cells. "Immune deserted" tumors show a total lack of lymphocytes at the invasive margin. Characteristics: Impaired extravasation of lymphocytes, CAF-S1 and hypoxia. "Inflamed-SR" tumors show lymphocytic infiltration in the stroma, but not intratumorally. Characteristics: PD-1 expression and CD69 on T cells, an increase in macrophages compared to the poorly infiltrated tumors, and neutrophil infiltration with IL-17 expression. "Inflamed-SI" tumors show lymphocytic infiltration in the stroma and intratumorally. Characteristics: High macrophage levels with an activated phenotype (CD206+), inflammatory molecules including GZMB and IFNγ. T cell expression of PD-1, CTLA-4, TIGIT and GITR, CD69 and CD103 (T_RM_). The question mark between immune-excluded and inflamed-SR emphasizes the uncertainty whether these are actual biological distinct phenotypes or are similar but with a different scale or magnitude of infiltration.

**Figure 2 cancers-13-00316-f002:**
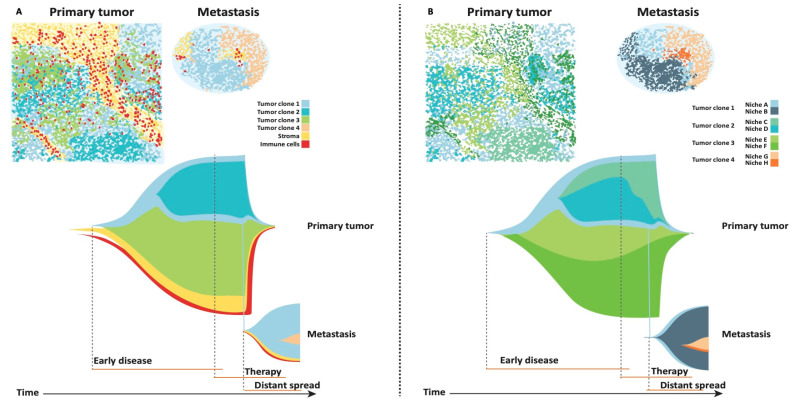
Evolution of tumor cells and the tumor micro-environment (TME) in TNBC. (**A**) An inflamed-SI tumor with four tumor clones, stroma, and immune cells illustrated. The evolution of the compartments is visualized below (evolution figure adapted from Sun, Zheng, and Curtis, 2018) and spans from early disease, during therapy to distant spread. The primary tumor consists of 3 tumor clones. The clones are surrounded by stroma and immune cells. Typically, these compartments are described separate since methods to reconstruct tumor evolution are based on genomic heterogeneity amongst tumor subclones and lack direct information regarding immune evolution. (**B**) Here, the same tumor is illustrated, however tumor subclones and the surrounding stroma and immune cells are instead clustered into ecological niches, revealing their topography within the tumor-immune microenvironment. Such spatially-resolved cellular maps may provide insight into why some subclones expand, whereas others are spatially restricted, thereby informing patterns of metastatic "spread" and "therapy response" in TNBC and will be enabled by scalable spatial genomics techniques. Spread: Clone 1 disseminates to distant sites and form a macro metastatic lesion. Under (Niche A) clone 1 can be found in both the primary tumor and metastasis. Under (Niche B) the same tumor clone ("1") escapes but now we can observe that the tumor microenvironmental niche in the metastasis changes quickly (from Niche A in light grey, to Niche B in dark grey), indicating that the clone is the same but the micro-environment drastically changes, potentially leading to a more aggressive phenotype. Therapy response: In the primary tumor clone 2 and 3 both show different microenvironmental niches, which respond differently to, e.g., therapy. For example, tumor clone 2 (Panel A) is divided in Niche C and D (Panel B), where the niches show differential response to therapy. Niche D rapidly decreases during therapy, whilst Niche C is less sensitive and persists longer. Consideration of the tumor-immune microenvironment during tumor progression may enhance our understanding of the evolutionary dynamics and drivers of tumor evolution.

**Table 1 cancers-13-00316-t001:** Methods to generate high-dimensional phenotypic data.

Technique	Summary	Modality	Spatial	Resolution	References
**Single cell RNA sequencing**	Single cell transcriptome sequencing to assess gene expression patterns for each cell individually	RNA	No	Single cell	[6,21,23,28,29]
**Spatial transcriptomics**	Spatial information is obtained by integrating imaging and positional barcoding.	RNA	Yes	~100s of cells	[30,31,32]
**TCR sequencing**	Single T cell receptor sequencing to profile the repertoire of T cell receptors	TCR sequence (clonotype)	No	Single cell	
**Flow cytometry**	Single cell labeling with fluorescent-tagged antibodies (~4 to 5 plex)	Protein	No	Single cell	[20,33,34]
**CyTOF**	Single cell labeling with metal-tagged antibodies (~40-plex) measured using laser ablation and mass spectrometry-based time-of-flight	Protein	No	Single cell	[19]
**Nanostring Digital Spatial Profiling**	Photocleavable oligonucleotide barcodes covalently linked to in-situ affinity reagents (antibodies/RNA probes)	Protein/RNA	Yes	~100s to 1000s of cells	[35,36,37]
**Multiplex immune-fluorescence**	Immunofluorescence with multiple antibodies (~4 to 5) to assess marker relationships in tissue	Protein	Yes	Single cell	[25,27,38]
**Imaging Mass Cytometry (IMC)**	Immunohistochemistry staining using metal metal-tagged antibodies (~40-plex) with laser ablation and mass spectrometry-based time-of-flight detection at cellular resolution in tissue	Protein	Yes	Single cell	[26,39]
**Multiplex ion beam imaging (MIBI)**	Multiplexed ion beam imaging by time of flight, uses bright ion sources and orthogonal time-of-flight mass spectrometry to image metal-tagged antibodies (~40-plex) at subcellular resolution in tissue	Protein	Yes	Single cell	[24,40]

## Data Availability

Data sharing not applicable.

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
