# Peer review of "A High-Dimensional Window into the Micro-Environment of Triple Negative Breast Cancer"

_cancers, 2021, doi:10.3390/cancers13020316_

Round 1
Reviewer 1 Report
This review focus on the interaction between triple negative breast cancer and the tissue microenvironment (TME). While genomic heterogeneity and subclonal diversity are prevalent in this subgroup of tumors, a growing body of evidence indicates that the disease course depends on the interaction between cancer cells and the tumor micro-environment (TME). Efforts to further elucidate the TME have been aided by a mix of new technologies that study tumors in a high-dimensional manner. These high-dimensional technologies enable comprehensive analysis of cell phenotypes at the single cell level or the spatial relationships of tumor and immune cells. In this review, they discuss the cellular phenotypes and spatial patterns of the lymphoid-, myeloid- and stromal cells in the TNBC microenvironment and the potential value of mapping these features onto tumor cell genotypes.
The introduction nicely describes the necessity of why one should focus on triple negative breast cancer. Targeted therapy for this subtype is lacking and prognosis is worse than the other subtypes. The interaction with the TME and the immune cells is emphasized. Spatial patterns of cells in the breast TME and single cell phenotypes is emphasized and the different methods of studying the individual cell types and appearance is nicely described. They focus on the phenotypes and patterns of the lymphoid cells, myeloid cells, and stromal cells in the local breast cancer TME.
Table 1 is a nice demonstration of methods to generate high-dimensional phenotypic data.
Figure 1 is illustrative and nice demonstrating the four spatial lymphocytic phenotypes of immune infiltration in TNBC.
Conclusion
High dimensional technologies have yielded fresh insights into the local TME within TNBC. The spatial composition of the TME in breast cancer is often expressed as one of four spatial phenotypes, depending on the localization of lymphoid immune cells. They emphasize the need of further research in the field. They end with the optimal goal; high-dimensional technologies will be instrumental in the quest to discover biomarkers to select patients for therapy and to uncover (targetable) weaknesses of TNBC. This is a major field and problematic issue for the clinicians and needs to be optimized to improve breast cancer treatment and survival.
Conclusively, the text includes many elements. Maybe too many to get the full overview. There are many cell types and interactions described. Maybe a table or a figure with the relevant cells and the relevant markers are illustrated
Author Response
We thank the reviewer for the feedback on our manuscript entitled “A high-dimensional window into the micro-environment of triple negative breast cancer”, and we appreciate the nice summary of the work and the suggested improvements. We have addressed the feedback in the requested 'point-by-point' structure:
- Comment 1 of reviewer: “Conclusively, the text includes many elements. Maybe too many to get the full overview. There are many cell types and interactions described. Maybe a table or a figure with the relevant cells and the relevant markers are illustrated”
Answer to comment 1: We agree that there is a lot of information in the review that can make it difficult to read. We therefore have added a section A to figure 1 which now explains the different cell types (with markers on their membranes) and how they interact with the stroma and other cells, as a summary.
Reviewer 2 Report
The review by Nederlof et al. entitled “A high-dimensional window into the microenvironment of triple negative breast cancer” proposes to summarize efforts to characterize the cell phenotypes present within the TNBC tumor microenvironment using high-dimensional analysis. In general the review was easy to follow and provided a good overview. The authors should carefully perform a thorough language edit before final submission.
The interpretation of Figure 1 is not obvious. It is not clear how the cell phenotypes are related to the stroma -perhaps using additional figures would help rather than just this one summary figure.
The review would benefit by adding additional commentary on the observations cited rather than merely listing the observations – the first half of the review does a nice job of providing critical insight, but sections 3.1-3.5 do not provide this insight.
It is not clear in sections 3.1-3.5 and 5.2 if these data are relevant to all breast cancers or to TNBC in particular. It is not a problem, but it should be clearly stated.
Author Response
We thank the reviewer for the feedback on our manuscript entitled “A high-dimensional window into the micro-environment of triple negative breast cancer” and appreciate the suggested improvements. We have addressed the feedback in the requested 'point-by-point' structure:
- Comment 1 of the reviewer: “The interpretation of Figure 1 is not obvious. It is not clear how the cell phenotypes are related to the stroma -perhaps using additional figures would help rather than just this one summary figure. “
Answer to comment 1: We have adjusted ‘Figure 1’ and included an additional illustration (A) to clarify cell phenotypes and how these are related to the tumor and stroma.
- Comment 2 of the reviewer: “The review would benefit by adding additional commentary on the observations cited rather than merely listing the observations – the first half of the review does a nice job of providing critical insight, but sections 3.1-3.5 do not provide this insight.”
Answer to comment 2:
We have provided adding additional commentary on the observations in 3.1-3.5, namely:
- Line 216-222: The phenotype of TRM and their transition in late dysfunctional profiles make them intriguing candidates to exploit in the anti-cancer immune response of patients with breast cancer. Indeed, the presence of TRM cells in the breast cancer TME (assessed by a single cell RNA derived signature) was associated with favorable overall and relapse-free survival in primary TNBC [23], as well as with response to pembrolizumab in patients with metastatic TNBC (KEYNOTE-086) [58]. However, to the best of our knowledge, preselection of breast cancer patients with TRM immune profiles is currently not incorporated in any clinical trial with (immune) therapeutical interventions.
- Line 232: Provided that the different phenotypes of CD4+ T cells are linked to a distinct function and interactions with cell types, distinct spatial patterns would be expected. scRNAseq of CD4+ T cells from several breast cancer subtypes reported that CD4+TEM and TCM clusters exhibit variable levels of gene expression involved in type I and II interferon response, hypoxia and anergy, indeed indicating a different signaling and role in the tumor [21]. However, there is only scarce information on the spatial position of CD4+ T cells and their clinical consequences.
- Line 268: It is currently unknown if the B cells infiltrated between the breast tumor cells have a significant different phenotype or function as opposed to the B cells in the TLS around the breast tumor. B cells’ diverse roles in humoral immunity, antigen presentation, modulation of T cells and innate immune cells [68] suggests there are various biological relevant spatial patterns and colocalizations with different cell types that yet have to be uncovered.
- Line 276: The potential of using immunoglobulin repertoires to study the anti-tumor response or to use as biomarkers to predict the efficacy of (immuno)therapeutic interventions [68] can be uncovered with future single cell RNA-based immunoglobulin repertoire studies.
- Line 314: Moreover, the differences or similarities of NK cells between TNBC and other breast cancer subtypes is still unknown. With the blocking of the inhibitory NKG2A receptor in head and neck cancer as a prelude [78,79], new results on targeting and exploiting NK cell activity in breast cancer trials (e.g. CT04307329) are expected.
Reviewer 3 Report
Overall, the manuscript by Nederlof et al. seems to me to be a good work of revision and of state-of-art on a complex and very current topic whose understanding could open new therapeutic perspectives for the treatment of many types of neoplasm by passing through insights of new facets of cancer biology. Although still limited, the data we have available in this field of oncological research undoubtedly suggest the importance and role of TME not only in the progression of BC disease but also in the potential therapeutic response. In this context, the authors' work is certainly relevant and the collection of information on the tumoral and non-tumoral cellular phenotypes and spatial topographies of TME in TNBC is as exhaustive as all the bibliographic mentions. If I have to indicate an aspect that is not exactly convincing and that could further improve the review, this is represented by the excessive technicality of some passages that could make you lose sight of the considerable biomedical fallout the understanding of these biological aspects could have. I realize the topic as really complex in itself, as well as all the technologies allowing for its investigation; therefore, rather than modifying the basic approach and therefore in accordance with the authors' work, I would suggest implementing the final sections with some "more practical" aspect the study of TME could contribute, above all in the development of new therapeutic strategies for BC and/or in the improvement of the existing ones on the basis of original knowledge. So, I propose to include more correlations between therapy and TME analysis. On balance, the authors are engaged in the development of immunotherapy by combination with conventional treatments as certain chemotherapies and targeted therapies. How could elucidation of TME improve the development of new effective (immuno)therapeutic options? Expand this point, conceivably by inserting a special section.
Minor points
- I would not use the future time to introduce new sections and topics. Hence, starting from line 72, I would delete "will" and I directly say "we discuss". Likewise, for line 143, 270, 430 and so on. Please check through the entire manuscript.
- Please verify the editing of Tab 1.
- Line 280, “indeed” at the beginning of the sentence.
- Line 285, delete “expression”.
- Line 310, delete starting from “,both…”
- Line 474, interactions.
- Line 471 and 475, recurrence of “However”. For example, at the line 475, I would start with “In this frame”.
Author Response
We thank the reviewer for the feedback on our manuscript entitled “A high-dimensional window into the micro-environment of triple negative breast cancer” and appreciate the suggested improvements. We have addressed the feedback in the requested 'point-by-point' structure:
- Comment 1 of the reviewer: “I realize the topic as really complex in itself, as well as all the technologies allowing for its investigation; therefore, rather than modifying the basic approach and therefore in accordance with the authors' work, I would suggest implementing the final sections with some "more practical" aspect the study of TME could contribute, above all in the development of new therapeutic strategies for BC and/or in the improvement of the existing ones on the basis of original knowledge. So, I propose to include more correlations between therapy and TME analysis.”
Answer to comment 1: We have implemented a final section on the clinical implication of TME profiling (Line 517-536). Here we have discussed how our current knowledge and the high-dimensional techniques could help to identify new therapeutic targets or patient groups. We hope that this additional section provides a useful link between therapy and TME analysis.
- Comment 2 of the reviewer: “If I have to indicate an aspect that is not exactly convincing and that could further improve the review, this is represented by the excessive technicality of some passages that could make you lose sight of the considerable biomedical fallout the understanding of these biological aspects could have.”
Answer to comment 2 of the reviewer: As the feedback included the notification of the exhaustive tumoral and non-tumoral cellular phenotypes and spatial topographies of TME in TNBC. To help summarize the different cellular phenotypes for readers, we have added a section A to figure 1.
- Textual comments of the reviewer:
- I would not use the future time to introduce new sections and topics. Hence, starting from line 72, I would delete "will" and I directly say "we discuss". Likewise, for line 143, 270, 430 and so on. Please check through the entire manuscript.
- Please verify the editing of Tab 1.
- Line 280, “indeed” at the beginning of the sentence.
- Line 285, delete “expression”.
- Line 310, delete starting from “,both…”
- Line 474, interactions.
- Line 471 and 475, recurrence of “However”. For example, at the line 475, I would start with “In this frame”.
Answer to textual comments of the reviewer: We have adjusted all textual suggestions that were kindly provided by the reviewer.
Round 2
Reviewer 3 Report
Based on the first review process, the authors further improved the quality and information present in the manuscript. In this way the initially highlighted points of criticism have been overcome and the review is in my opinion ready for publication.